# Multi-Component Characterization and Quality Evaluation Strategy of *Sarcandrae* Herba by Combining Dual-Column Tandem HPLC Fingerprint and UPLC-Q-TOF-MS/MS

**DOI:** 10.3390/molecules30081825

**Published:** 2025-04-18

**Authors:** Zhijian Zhong, Pan Deng, Xiaorong Luo, Weifeng Zhu, Pengdi Cui, Zhe Li, Zhiqiang Xiao, Yu Shen, Xinyu Wu

**Affiliations:** 1Jiangzhong Pharmaceutical Co., Ltd., Nanchang 330100, China; zzj@crjz.com (Z.Z.); dengpan@crjz.com (P.D.); lxr@crjz.com (X.L.); cuipengdi@crjz.com (P.C.); xzqiang@crjz.com (Z.X.); wuxinyu@crjz.com (X.W.); 2Technology and Innovation Center of Jiangxi Traditional Chinese Medicine Manufacturing and Process Quality Control, Nanchang 330004, China; 3Key Laboratory of Modern Preparation of TCM, Ministry of Education, Jiangxi University of Chinese Medicine, Nanchang 330004, China; lizhezd@163.com (Z.L.); shenyu240568@163.com (Y.S.)

**Keywords:** *Sarcandrae* herba, fingerprint, mass spectrometry, material basis, identification of origin

## Abstract

A dual-column tandem mode was used to establish the fingerprints of *Sarcandrae* herb from different origins, and their chemical compositions were characterized by UPLC-Q-TOF-MS/MS, which provided an experimental basis for the establishment of a rapid and efficient method for the overall quality control of *Sarcandrae* herba. For the first time, nine common components were identified from the *Sarcandrae* herba herbs of 24 origins, which were neochlorogenic acid, chlorogenic acid, 4-caffeoylquinic acid, eleutheroside B1, quercetin-3-O-β-D-glucuronide, neoastilbin, astilbin, isofraxidin, and rosmarinic acid, respectively. A total of 92 compounds were identified by liquid mass spectrometry. The quality of the *Sarcandrae* herb from 24 origins was analyzed by similarity evaluation, principal component analysis, and cluster analysis, and the chemical components of *Sarcandrae* herba were identified by UPLC-Q-TOF-MS/MS. The results showed that the overall analysis based on fingerprinting and mass spectrometry could differentiate the origins of the herbs.

## 1. Introduction

*Sarcandrae* herba (SH) is the dried whole grass of *Sarcandra glabra* (Thunb.) Nakai, a plant of the genus *Sarcandra* in the family of *Araceae*, which is mainly produced in China, and also has a large distribution in eight countries, including Vietnam, Japan, Korea, Cambodia, Malaysia, Philippines, and India [1,2]. The herb has abundant chemical constituents, many biological activities, and wide range of pharmacological effects [3,4,5,6,7], and it is used in many kinds of Chinese medicinal preparations, such as compounded *Sarcandra glabra* tablets [8], Xuekang oral liquid, and Xinhuang tablets [9], etc. Therefore, it is necessary to establish a complete quality evaluation system for SH herbs.

The quality of traditional Chinese medicines (TCMs) has become a serious constraint to the development of the Chinese medicine industry due to the confusion of the origin of TCMs and seed sources [10,11]. The Pharmacopoeia of the People’s Republic of China is the main quality testing basis for all kinds of TCMs in China at present [12], and its quality evaluation indexes mainly include general routine contents such as traits, microscopic and thin-layer identification, etc. In addition, it also draws on the way of chemical medicines, selecting one or two chemical components for content determination. The information of these testing indexes stipulated in the Pharmacopoeia is not comprehensive enough and does not fully reflect the whole of the herbs; in particular, it is difficult to effectively distinguish the differences in the origin of the herbs, which also directly affects the standardized quality control of TCMs [13].

It is a common practice in current research to focus on a single chemical component as an indicator for quality control, but this model is not in line with the multi-component, multi-targeted nature of TCMs. The overall quality control of TCMs is more in agreement with the characteristics of the complex system of TCM. In the past decade, researchers have adopted high-throughput mass spectrometry (MS) technology to comprehensively analyze its chemical composition [14,15]. Fingerprinting (characterization) of TCMs, as an accurate and operable technology that can reflect the overall quality of TCMs to a certain extent, has been developed rapidly over the years [16]. Its advantage lies in the fact that it reflects the overall quality of TCMs as comprehensively and vaguely as possible at the present stage when most of the active ingredients are not clearly defined, which is in line with the characteristics of the complex system of TCM [17].

Wang et al. [18] used high-performance liquid chromatography fingerprinting combined with chemometrics, including cluster analysis (CA), principal component analysis (PCA), and orthogonal partial least squares discriminant analysis (OPLS-DA), to analyze and compare the genuine *Clematidis armandii* Caulis and related adulterated products. Jiménez-Carvelo [19] developed a new analytical method for the differentiation of olive oil from other vegetable oils using reversed-phase liquid chromatography and chemometric techniques. Chromatographic fingerprints of the methyl ester exchange sites of each vegetable oil were obtained using a short 3 cm column and evaluated in combination with a multivariate classification method to differentiate olive oil from other vegetable oils. Fraige et al. [20] investigated the anthocyanin profiles of 11 different grape cultivars and origins, and identified 20 anthocyanins by absorbance spectroscopy and fragmentation patterns in tandem mass spectrometry. A multivariate approach introducing PCA was used to assess the differences between cultivars. The results showed that the main reason for the isolation of hybrid grapes from the group of cultivars represented by grapes was the anthocyanin diglucoside. The study was only able to differentiate between the different cultivars of grapes, while their origin could not be effectively recognized.

In this study, based on the collection of representative samples of SH from various origins, the fingerprint and MS technique were applied to carry out the source study of SH, and the dual-column tandem SH fingerprints were established and analyzed for their composition. Combined with MS analysis and reference compounds, the common components in SH from different origins were identified. Using PCA and cluster analysis, statistical analysis was conducted on the common peaks of the fingerprints and the full spectrum of the fingerprints and MS to explore effective identification methods for distinguishing the origin of SH medicinal herbs.

## 2. Results and Discussion

### 2.1. Optimization of Fingerprinting Method

The traditional liquid chromatographic separation process usually selects a single column for HPLC separation, but the separation of components is not effective [21]. In this study, the results of the HPLC analysis of the components of the herbs from different origins are shown in Figure 1. Compared with the single-column mode, the dual-column tandem mode can effectively realize the baseline separation of compounds, eliminate the influence of interference components in the actual sample to a certain extent, and effectively improve the separation and analysis of complex samples. The separation effect of the components in the dual-column tandem mode was significantly better than that in the single-column HPLC separation process, and the dual-column tandem HPLC separation mode realized the great separation of the components of the herbs [22]. The methodology of dual-column tandem mode HPLC was validated, and its precision, reproducibility, and stability could meet the requirements of fingerprinting (six repetitive injections), and the relative retention time (RSD) of each common peak was less than 0.80%, and the relative peak area RSD of each common peak was less than 2.50%, and the fingerprints of 24 samples are shown in Figure 2.

### 2.2. Common Components in SH Medicinal Herbs from 24 Different Origins

Comparing the fingerprints of 24 herbs of different origins, 12 peaks were identified as the common fingerprint peaks of SH herbs. For the identification of the 12 common components, MS analysis was performed on SH herbs from six provinces, and the results of the MS analysis were identified by means of database and reference standard comparisons. The raw MS data obtained were imported into PeakView 1.2 software to obtain the total ion current maps in positive and negative ion modes (Figure 3). The target compounds were screened by the XIC Manager function of PeakView software, and the primary mass spectral data and secondary spectral fragment ions of the compounds were analyzed comprehensively to identify and confirm the target compounds. The XIC Manager function in PeakView software was used to screen the target compounds, and the primary mass spectrometry data and secondary spectra of the compounds were combined with the fragment ions and other information for comprehensive analysis, so as to realize the identification of the target compounds. A total of 92 compounds were identified from the herbs, and the specific data are shown in Table 1. Combined with the MS data and reference standard comparison, the common peaks 1, 2, 3, 4, 6, 7, 8, 9, and 11 were identified as neochlorogenic acid, chlorogenic acid, cryptochlorogenic acid, eleutheroside B1, quercetin-3-O-glucuronide, neoastilbin, astilbin, isofraxidin, and rosmarinic acid, respectively, as shown in Figure 4.

The test results of the samples of each origin were imported into the similarity evaluation system software Chinese Medicine Chromatographic Fingerprint Similarity Evaluation System (Version 2012.130723), and the sample S1 was selected as the reference fingerprint, and the median method was used as the method of generating the control fingerprint, and the multi-point correction method was combined to match the liquid phase profiles of each origin. The results of similarity evaluation are shown in Table 2. The average similarity of SH samples from 24 origins was 0.926, indicating that the chromatographic patterns of SH samples from different origins are relatively similar, and their origins need to be further mined using machine learning algorithms for identification.

### 2.3. Fingerprint-Based Origin Identification of SH Herbs

Principal components were introduced to analyze the differences between the common peaks in the fingerprints of SH herbs from different origins. The results are presented in the form of a scatter plot of principal component scores about each sample, which are shown in Figure 5. The accumulated variation of the first two principal components reached 66.30% using the first two principal components as the horizontal and vertical coordinates of the scatterplot, respectively. Each point in the scatterplot represents an SH herb origin, and its degree of aggregation reflects the similarity of SH herbs. It can be seen that in the results of principal component analysis based on the common peaks, the differences of SH herbs from various origins were not substantial, and the sample points from other origins were basically clustered in one region except for Fujian province and Sichuan province.

The insignificant effect of origin differentiation based on common peaks suggests that these components may be the reason why SH herbs from different origins all exert antimicrobial and antiviral therapeutic effects. Among them, chlorogenic acid, rosmarinic acid, neochlorogenic acid, and 4-caffeoylquinic acid are some of the most prominent medicinal efficacy components, most of which have antibacterial, anti-inflammatory, anti-platelet aggregation, anti-toxicity, and other biological activities [71,72,73]. Liu et al. [74] found that the phenolic acid components of organic acids, such as neochlorogenic acid, chlorogenic acid, 4-caffeoylquinic acid, caffeic acid, and rosmarinic acid, were the main components in *Sarcandra glabra* injection by HPLC analysis, and the anti-inflammatory effect of *Sarcandra glabra* injection was confirmed by in vitro and ex vivo pharmacological experiments. In vivo experimental studies revealed that rosmarinic acid in SH could reduce the mortality rate of mice with pneumonia caused by influenza virus infection by down-regulating the secretion of gamma-interferon (IFN-gamma) and tumor necrosis factor-alpha (TNF-alpha) by helper T-cell 1 (Th1) and up-regulating the secretion of interleukin-4 (IL-4) and IL-5 by T-cell 2 (Th2) [7]. Flavonoids, rosmarinic acid, and astilbin in SH play an antioxidant role by directly and indirectly scavenging ROS (e.g., Fe^2+^ chelation), and their ROS scavenging may be based on hydrogen-atom transfer or an electron-transfer pathway [73]. As a critical indicator for quality control of SH, isofraxidin attenuated the IL-1β-induced significant increase in inflammatory mediators and cytokines such as nitric oxide (NO), inducible NO synthase (iNOS), cyclooxygenase-2 (COX-2), prostaglandin E2 (PGE2), tumor necrosis factor alpha (TNF-alpha), and IL-6, and also inhibited the induction of matrix metalloproteinase (MMP)-3 and MMP-13 by IL-1β [75].

PCA of the full spectrum of fingerprints of SH herbs from 24 origins showed that the variations represented by the first and second principal components reached 85.21% and 11.27%, respectively (Figure 6). It can be seen that the 24 samples from six provinces were well differentiated, and the SH herbs from different provinces had exclusive distribution areas on the principal component score plot, which was quite different from the results of principal component analysis based on fingerprint common peaks. The results showed that HPLC fingerprints can comprehensively characterize the overall quality of TCMs through the overall analysis mode, and provide technical support for the construction of a comprehensive quality evaluation system based on multi-indicator components, which is of significant application value for the objective identification of the quality grade of herbal medicines and the traceability of authenticity.

### 2.4. MS-Based Analysis of the Material Basis of SH Herbs

As a further means of comparing the differences between the components in SH samples from different origins, the sample information is presented in Figure 7 as a fingerprint, which normalizes the signal peaks for each substance from the different origins, transforming them into the range from 0 to 1. The fingerprint contains all the signals that can be detected by the instrument, where each row represents a sample and each column represents a substance. In Figure 7, the substances in regions I, II, III, IV, and V represent organic acids, sesquiterpenoids, flavones, coumarins, and other classes of characteristic components, respectively, and the differences in compositional composition of SH herbs from different origins can be observed distinctly. The content of organic acids, which play a major role in medicinal effects, was significantly higher in SH from Yunnan than in other origins, sesquiterpenoids accounted for a larger proportion of SH from Jiangxi, SH from Guangxi appeared to be dominated by flavones, and coumarins were more prevalent in SH from Yunnan and Guizhou.

The total ion current spectra of the MS of SH herbs from six provinces were analyzed by systematic clustering using the intergroup linkage method, and the Euclidean distance was used for the calculation of inter-sample distance, and the results are shown in Figure 8. It can be found that the calculation results verified the results of the fingerprint-based full-spectrum PCA analysis from another perspective, and there was a more powerful similarity between the SH from Guangxi and Sichuan, as well as between the SH from Sichuan and Fujian. This further indicates that there exists a significant origin specificity in the seed source of SH, and also points out the direction for the study of the efficacy of SH from different origins in a later study.

## 3. Materials and Methods

### 3.1. Sources and Preparation of Samples

Twenty-four batches of SH of different origins were purchased from Jiangxi Jiangzhong Prepared Slices of Chinese Crude Drugs Co., Ltd. (Jiujiang, China). The samples are specifically shown in Table 3. All the samples were identified as the dried whole herb of *Sarcandra glabra* (Thunb.) Nakai from the family of *Araceae*, by Dr. Liu Yong, Jiangxi University of Traditional Chinese Medicine. Representative samples were deposited at the Advanced Manufacturing Research Laboratory, Huarun Jiangzhong Pharmaceutical Group Co., Ltd., China.

SH herbs were pulverized using a grinder (Shanghai Filiberto Food Machinery Co., Ltd., Shanghai, China) and then sieved through a sieve (particle size < 0.355 mm). For the extracts, each powdered sample was accurately weighed 1.0 g using an electronic balance (Sartorius Scientific Instruments Co. Ltd., Beijing, China), dispersed in methanol (60%, 50 mL), weighed, and then ultrasonicated (250 W, 35 KHZ) for 30 min. The extracts were cooled to room temperature and weighed again; 60% methanol was added to maintain the same weight as before extraction, mixed well, and filtered. The renewed filtrate was taken and filtered through 0.22 μm organic microporous filter membrane.

### 3.2. Chemicals and Reagents

Twenty reference compounds were used: isofraxidin (Lot No. 110837-202009), chlorogenic acid (Lot No. 110753-202119), caffeic acid (Lot No. 110885-201703), rosmarinic acid (Lot No. 111871-202007), atractylenolide II (Lot No. 111876-201501), rutin (Lot No. 100080-202012), fraxetin (Lot No. 111731-202103), quercetin (Lot No. 100081-201610), astilbin (Lot No. 111798-202306), linolenic acid (Lot No. 111631-202207), linoleic acid (Lot No. 111622-202105), purchased from National Institutes for Food And Drug Control; 7-hydroxycoumarin (Lot No. 111739-200501) purchased from National Institute for the Control of Pharmaceutical and Biological Products (Shanghai, China); coumarin (Lot No. 137-12-10) purchased from Guangzhou Jiatu Technology Co., Ltd. (Guangzhou, China); quercetin-3-O-β-D-glucuronide (Lot No. ST87701) purchased from Shanghai Shidande Standard Technical Service Co., Ltd. (Shanghai, China); kaempferol 3-O-β-D-glucuronide (Lot No. CFN90359) purchased from Wuhan Tianzhi Biotechnology Co., Ltd. (Guangzhou, China); eleutheroside B1 (Lot No. O27GB165514), neoastilbin (Lot No. O17HB198038), kaempferol (Lot No. J17IB218678) purchased from Shanghai Yuanye Biotech. Co., Ltd. (Shanghai, China); neochlorogenic acid (Lot No. 50190020), 4-caffeoylquinic acid (Lot No. 63180010) purchased from Shanghai Anpu Experimental Technology Co., Ltd. (Shanghai, China); the purity of all reference compounds was greater than 98%. The experimental reagents were mass spectrometry grade methanol (Supelco Inc., Bellefonte, PA, USA), acetonitrile (Merck KGaA, Darmstadt, Germany), and methanoic acid (Thermo Fisher Scientific Inc., Waltham, MA, USA).

### 3.3. Standard Solutions

The appropriate amount of each reference substance was weighed precisely, placed in a 50 mL brown volumetric flask, added with methanol, and dissolved by ultrasonication and volume, and then passed through 0.22 µm microporous filter membrane to obtain the standard solutions.

### 3.4. Analytical Conditions for Fingerprinting

The fingerprints of the herbs were established by dual-column tandem high-performance liquid chromatography (HPLC), in which a high-performance liquid chromatograph (Waters e2695, Waters Corporation, Milford, MA, USA) was equipped with a pre-column Agilent 5HC-C18 (250 mm × 4.6 mm, 5 µm) and a post-column Eclipse Plus Phenyl-Hexyl (250 mm × 4.6 mm, 5 µm). The column temperature was 30 °C, the detection wavelength was 330 nm, and the injection volume was 10 µL. The elution solvents were 0.1% formic acid acetonitrile solution (A)–0.1% formic acid aqueous solution (B), and the elution procedures were as follows: 0–15 min, 15% A–15% A; 15–60 min, 15% A–30% A; 60–90 min, 30% A–60% A. The volume flow rate of the elution solvent was 0.3 mL/min for the first 15 min, and 0.5 mL/min for the remaining time.

### 3.5. Parameters for MS Analysis

The assay was performed using an AB SCIEX Triple TOF 5600 liquid mass spectrometer with an electrospray ionization source, and the data were collected in positive and negative ion modes. Mass scan range (*m*/*z*): 100~1500, spray voltage: 5500 V, ion source temperature: 500 °C, de-clustering voltage: 100 V, collision energy: 45 V, auxiliary gas 1: 50 psi, auxiliary gas 2: 50 psi, curtain gas: 40 psi. The data collection time was 55 min, and the data were collected using the TOF-MS-IDA-MS/MS mode. In the IDA setup, the six highest peaks with a response value exceeding 100 cps were selected for secondary mass spectrometry scanning. Sub-ion scanning range *m*/*z*: 50 to 1250, and those that met the conditions were preferentially subjected to secondary scanning. Dynamic background deduction was turned on.

The constituents of SH were identified by reviewing the relevant literature with the use of the SciFinder, Reaxys, and ChemSpider databases. The XIC Manager target screening function of PeakView 1.2 software was adopted to preliminarily screen the components with mass deviation less than 5 × 10^−6^ based on the addition ion [M+H]^+^, and to obtain the secondary fragmentation information of eligible quasi-molecular ions, which was further compared with the cleavage pattern of the control and the literature.

## 4. Conclusions

In this study, 24 SH herbs of different origins were collected, HPLC fingerprints were established, and the compositional studies were conducted based on MS analysis, and a total of 92 components were identified, and the components of the nine common characteristic peaks among the 12 common peaks were confirmed in combination with the control. Nine common peaks were identified from the SH herbs of 24 origins, which were neochlorogenic acid, chlorogenic acid, 4-caffeoylquinic acid, eleutheroside B1, quercetin-3-O-β-D-glucuronide, neoastilbin, astilbin, isofraxidin, and rosmarinic acid, respectively. In view of the complexity of the information of SH fingerprints, principal component analysis and cluster analysis were introduced to compare the common and overall components of SH from different origins. The results showed that principal component analysis based on the common components could not realize the distinction of the origin of the herbs, while the integrated analysis based on fingerprint profiles and mass spectrometry could realize the distinction of the origin of the herbs. These findings demonstrate that the integrated application of chromatographic fingerprinting and MS provides a comprehensive strategy to characterize TCM as a complex system, effectively capturing its holistic quality attributes. The established methodology not only enables origin discrimination but also lays a foundation for future pharmacological investigations. Specifically, the identified characteristic components (e.g., chlorogenic acid, rosmarinic acid) could serve as key markers for correlating geo-authenticity with therapeutic efficacy. Further studies may focus on: (1) validating the bioactivity of these chemical markers through in vitro/vivo models to establish structure–activity relationships; (2) developing quality control protocols based on multi-component synergism to guide standardized production of SH herbs; (3) integrating multi-omics data (e.g., metabolomics, pharmacodynamics) to decipher the “multi-component, multi-target” mechanisms underlying TCM’s clinical effects. Such efforts will advance the translation of fingerprint-based differentiation into evidence-driven applications for precision herbology and sustainable utilization of medicinal resources.

## Figures and Tables

**Figure 1 molecules-30-01825-f001:**
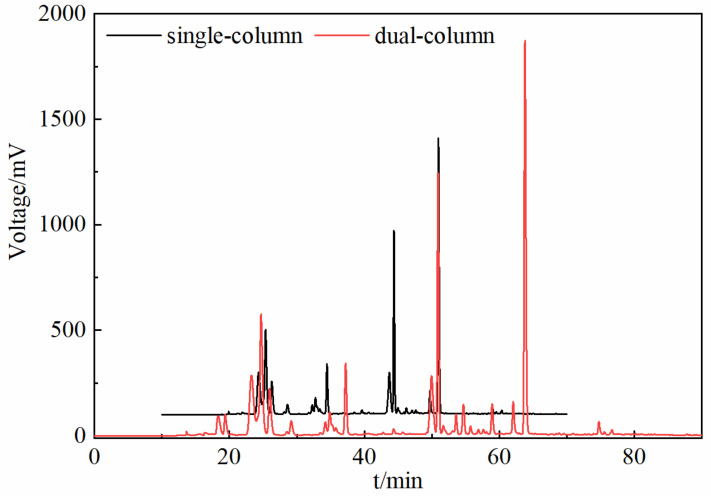
Comparison of separation results of two HPLC modes (—: single-column mode fingerprints; —: dual-column tandem mode fingerprints).

**Figure 2 molecules-30-01825-f002:**
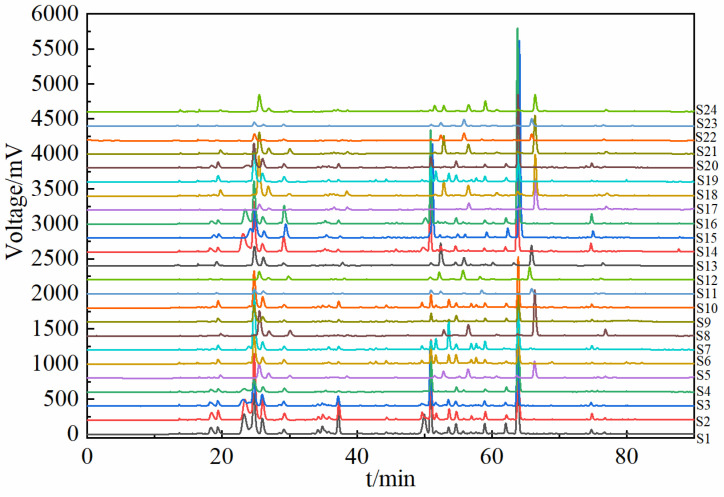
HPLC fingerprints of 24 batches of samples.

**Figure 3 molecules-30-01825-f003:**
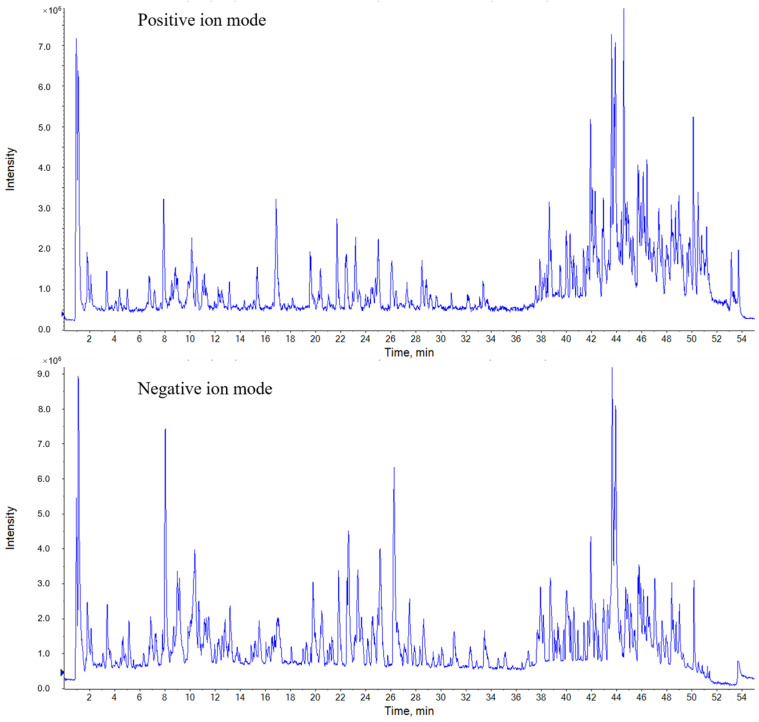
Total ion current diagrams of SH herbs in positive and negative ion mode.

**Figure 4 molecules-30-01825-f004:**
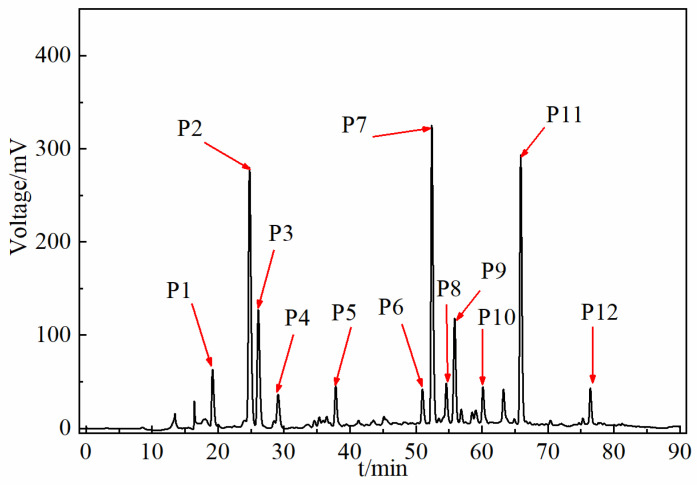
Reference fingerprint (characterization) of SH herbs (P1: neochlorogenic acid; P2: chlorogenic acid; P3: cryptochlorogenic acid; P4: eleutheroside B1; P6: quercetin-3-O-glucuronide; P7: neoastilbin; P8: astilbin; P9: isofraxidin; P11: rosmarinic acid).

**Figure 5 molecules-30-01825-f005:**
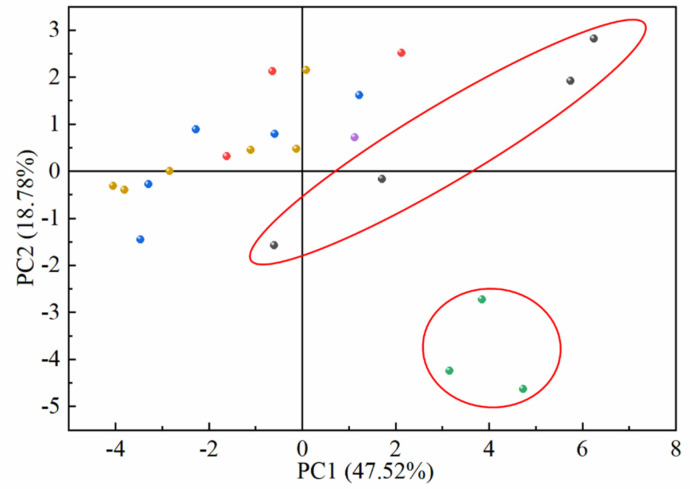
Origin differentiation based on shared components of fingerprints (●: Fujian; ●: Guizhou; ●: Jiangxi; ●: Sichuan; ●: Yunnan; ●: Guangxi).

**Figure 6 molecules-30-01825-f006:**
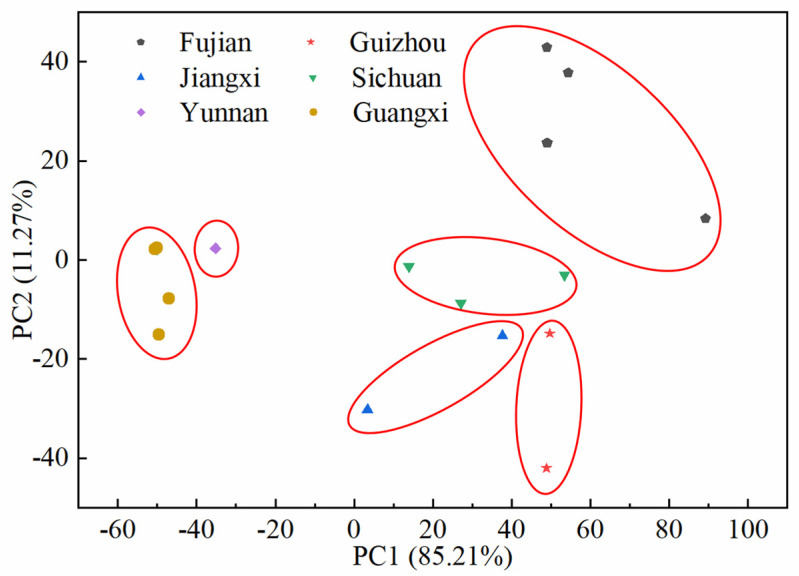
Distinction of origin based on the full spectrum of fingerprints.

**Figure 7 molecules-30-01825-f007:**
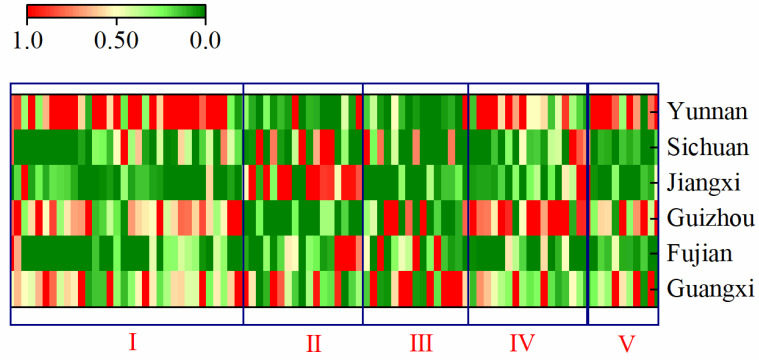
Each row in the figure represents all peaks in a sample; each column represents the peak of the same compounds in different samples.

**Figure 8 molecules-30-01825-f008:**
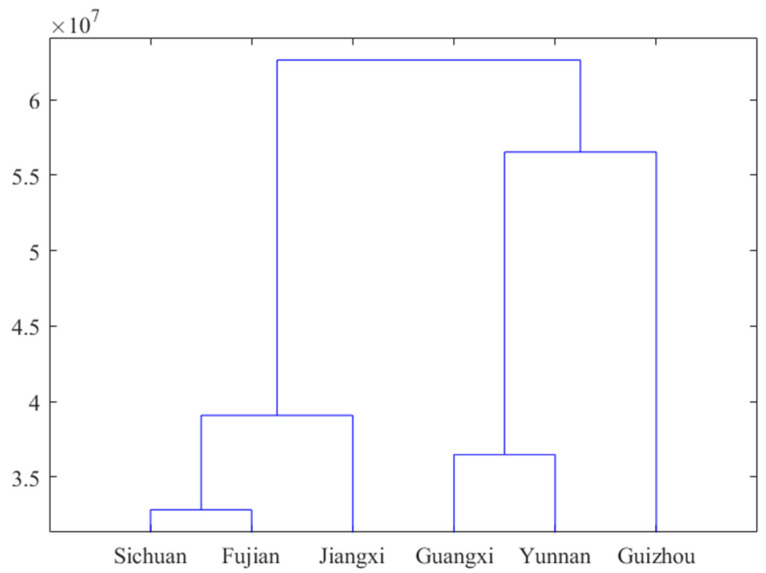
Clustering of SH herbs of different origins based on MS total ion mobility map.

**Table 1 molecules-30-01825-t001:** UPLC-Q-TOF-MS/MS analysis results of SH.

No.	RT (min)	Compound	Formula	Mode	*m*/*z*	Error (10^−6^)	Basis	Comment
1	1.10	Quinic acid	C_7_H_12_O_6_	^−^H	191.0565	2.2	[23]	organic acid
2	1.40	Shikimic acid	C_7_H_10_O_5_	^−^H	173.0459	2.0	[24]	organic acid
3	2.28	Gallic Acid	C_7_H_6_O_5_	^−^H	169.0149	3.8	[24]	organic acid
4	5.12	Neochlorogenic acid	C_16_H_18_O_9_	^−^H	353.0867	−3.2	RS [25]	organic acid
5	5.48	p-Coumaric acid	C_9_H_8_O_3_	^−^H	163.0409	4.8	[26]	organic acid
6	5.92	Esculin	C_15_H_16_O_9_	^+^H	341.0870	0.8	[27]	coumarin
7	7.19	1-p-Coumaroylquinic acid	C_16_H_18_O_8_	^−^H	337.0922	−2.1	[28]	organic acid
8	7.80	Catechin	C_15_H_14_O_6_	^−^H	289.0720	1.0	[29]	polyphenol
9	7.90	7-Hydroxycoumarin	C_9_H_6_O_3_	^+^H	163.0389	−0.5	[30]	coumarin
10	7.95	Scopolin	C_16_H_18_O_9_	^+^H	355.1022	−0.5	[31]	coumarin
11	8.03	Chlorogenic acid	C_16_H_18_O_9_	^−^H	353.0867	−3.0	RS [32]	organic acid
12	8.23	Esculetin	C_9_H_6_O_4_	^+^H	179.0336	−1.4	[33]	coumarin
13	8.88	Caffeic acid	C_9_H_8_O_4_	^−^H	179.0356	3.4	RS [23]	organic acid
14	8.90	3-Feruloylquinic acid	C_17_H_20_O_9_	^−^H	367.1021	−3.6	[34]	organic acid
15	8.99	4-Hydroxycoumarin	C_9_H_6_O_3_	^+^H	163.0389	−0.2	[33]	coumarin
16	9.16	4-Caffeoylquinic acid	C_16_H_18_O_9_	^−^H	353.0867	−3.1	RS [23]	organic acid
17	9.49	Fraxin	C_16_H_18_O_10_	^+^H	371.0981	2.3	[35]	coumarin
18	9.82	8-Hydroxy-6-methoxycoumarin	C_10_H_8_O_4_	^+^H	193.0493	−1.1	[36]	coumarin
19	10.12	Fraxetin	C_10_H_8_O_5_	^+^H	209.0441	−1.5	RS [37]	coumarin
20	10.28	Eleutheroside B1	C_17_H_20_O_10_	^−^H	383.0966	−4.6	RS [38]	coumarin
21	10.28	Fraxidin	C_11_H_10_O_5_	^−^H	221.0460	2.1	[39]	coumarin
22	10.68	Catechin 3-O-α-L-rhamnopyranoside	C_21_H_24_O_10_	^−^H	435.1295	−0.4	[40]	polyphenol
23	10.68	Epicatechin	C_15_H_14_O_6_	^−^H	289.0720	0.6	[40]	polyphenol
24	11.20	Benzyl alcohol xylopyranosyl-(1-6)-glucopyranoside	C_18_H_26_O_10_	^−^H	401.1434	−4.8		other
25	11.30	Coumarin	C_9_H_6_O_2_	^+^H	147.0440	−0.5	[27]	coumarin
26	11.46	5-p-Coumaroylquinic acid	C_16_H_18_O_8_	^−^H	337.0918	−3.2	[41]	organic acid
27	11.53	1-Caffeoylquinic acid	C_16_H_18_O_9_	^−^H	353.0881	0.8	[23]	organic acid
28	11.60	5-O-Caffeoylshikimic acid	C_16_H_16_O_8_	^−^H	335.0763	−2.8	[42]	organic acid
29	11.93	3-O-Caffeoylshikimic acid	C_16_H_16_O_8_	^−^H	335.0767	−1.5	[43]	organic acid
30	12.20	4-p-Coumaroylquinic acid	C_16_H_18_O_8_	^−^H	337.0921	−2.5	[41]	organic acid
31	12.35	Fraxidin-8-O-β-D-glucopyranoside	C_17_H_20_O_10_	^−^H	383.0985	0.2	[44]	coumarin
32	12.61	Kaempferol 3,7-diglucoside	C_27_H_30_O_16_	^−^H	609.1463	0.3	[26]	flavones
33	12.78	4-O-Caffeoylshikimic acid	C_16_H_16_O_8_	^−^H	335.0766	−1.9	[43]	organic acid
34	13.48	Ferulic acid	C_10_H_10_O_4_	^−^H	193.0511	4.8	[34]	organic acid
35	13.73	5-Feruloylquinic acid	C_17_H_20_O_9_	^−^H	367.1020	−4.1	[41]	organic acid
36	14.09	4-Feruloylquinic acid	C_17_H_20_O_9_	^−^H	367.1022	−3.4	[41]	organic acid
37	15.05	Scopoletin	C_10_H_8_O_4_	^+^H	193.0494	−0.8	[45]	coumarin
38	15.24	3-p-Coumaroylquinic acid	C_16_H_18_O_8_	^−^H	337.0917	−3.5	[41]	organic acid
39	15.35	Sarcaglaboside G	C_21_H_30_O_9_	^+^H	427.1965	0.7	[46]	sesquiterpenoids
40	17.03	Isofraxidin	C_11_H_10_O_5_	^−^H	221.0458	1.2	RS [47]	coumarin
41	17.12	8β,9α-Dihydroxyeudesman-4(15), 7(11)-dien-8α,12-olide	C_15_H_20_O_4_	^+^H	265.1433	−0.5	[48]	sesquiterpenoids
42	18.44	Liriope Muscari baily saponins C	C_21_H_20_O_11_	^−^H	447.0929	−1.0	[49]	flavones
43	19.29	Neoastilbin	C_21_H_22_O_11_	^−^H	449.1066	−5.2	RS [50]	flavones
44	19.58	Rutin	C_27_H_30_O_16_	^−^H	609.1465	0.6	RS [51]	flavones
45	19.72	Quercetin-7-O-β-D-glucopyranoside	C_21_H_20_O_12_	^−^H	463.0868	−3.0	[49]	flavones
46	19.79	8β,9α-Dihydroxylindan-4(5), 7(11)-dien-8α,12-olide	C_15_H_18_O_4_	^+^H	263.1277	−0.5	[52]	sesquiterpenoids
47	19.81	Quercetin-3-O-glucuronide	C_21_H_18_O_13_	^−^H	477.0654	−4.4	RS [53]	flavones
48	20.15	Isoquercitrin	C_21_H_20_O_12_	^−^H	463.0883	0.2	[49]	flavones
49	20.23	Chloranoside B	C_21_H_28_O_9_	^+^H	425.1810	0.9		sesquiterpenoids
50	20.47	Astilbin	C_21_H_22_O_11_	^−^H	449.1069	−4.5	RS [50]	flavones
51	20.53	Urolignoside	C_26_H_34_O_11_	^+^H	567.2084	2.0	[54]	other
52	21.08	1α,8α,9α-Trihydroxyeudesman-3(4),7(11)-dien-8β,12-olide	C_15_H_20_O_5_	^+^H	281.1381	−0.8		sesquiterpenoids
53	21.88	Atractylenolide IV	C_15_H_22_O_4_	^+^H	267.1591	0.0	[39]	sesquiterpenoids
54	22.21	Scoparone	C_11_H_10_O_4_	^+^H	207.0650	−0.8	[37]	coumarin
55	22.68	Rosmarinic acid 4-O-β-D-glucoside	C_24_H_26_O_13_	^−^H	521.1302	0.3	[55]	organic acid
56	22.89	Kaempferol 3-O-rutinoside	C_27_H_30_O_15_	^−^H	593.1517	0.9	[33]	flavones
57	23.08	Methyl caffeate	C_10_H_10_O_4_	^−^H	193.0512	3.1	[56]	organic acid
58	23.24	4,5-Dicaffeoylquinic acid	C_25_H_24_O_12_	^−^H	515.1175	−3.9	[23]	organic acid
59	23.37	Luteolin 7-O-glucuronide	C_21_H_18_O_12_	^−^H	461.0710	−3.4	RS [27]	flavones
60	23.52	Astragalin	C_21_H_20_O_11_	^−^H	447.0935	0.4	[49]	flavones
61	23.68	3,5-Dicaffeoylquinic acid	C_25_H_24_O_12_	^−^H	515.1172	−4.5	[23]	organic acid
62	24.89	Sarcaglaboside A	C_21_H_30_O_8_	^+^H	411.2010	−1.0	[46]	sesquiterpenoids
63	24.97	Glabraoside A	C_30_H_30_O_13_	^−^H	597.1620	1.0	[46]	other
64	26.27	Rosmarinic acid	C_18_H_16_O_8_	^−^H	359.0776	0.9	RS [55]	organic acid
65	26.27	Salvianic acid A	C_9_H_10_O_5_	^−^H	197.0464	4.3	[56]	organic acid
66	26.56	3,4-Dicaffeoylquinic acid	C_25_H_24_O_12_	^−^H	515.1173	−4.2	[23]	organic acid
67	27.49	3-p-Coumaroyl-5-caffeoylquinic acid	C_25_H_24_O_11_	^−^H	499.1250	0.8	[57]	organic acid
68	27.50	Phloridzin	C_21_H_24_O_10_	^−^H	435.1295	−0.3	[58]	flavones
69	28.49	Chloranoside A	C_21_H_28_O_9_	^+^H	425.1806	−0.1	[46]	sesquiterpenoids
70	29.35	N-trans-Feruloyltyramine	C_18_H_19_NO_4_	^−^H	312.1238	−1.1	[42]	organic acid
71	30.42	4-Caffeoyl-5-p-coumaroylquinic acid	C_25_H_24_O_11_	^−^H	499.1217	−5.9	[57]	organic acid
72	31.78	Quercetin	C_15_H_10_O_7_	^−^H	301.0347	−2.2	RS [49]	flavones
73	32.09	Istanbulin A	C_15_H_20_O_4_	^+^H	265.1432	−1.0	[59]	sesquiterpenoids
74	32.32	Methyl rosmarinate	C_19_H_18_O_8_	^−^H	373.0919	−2.8	[60]	organic acid
75	33.60	Sarcandralactone B	C_15_H_20_O_3_	^+^H	249.1483	−0.7	[61]	sesquiterpenoids
76	35.07	1,4-Dioxaspiro[4.4]nonane-6-heptanoic acid, 8-(acetyloxy)-7-carboxy-, 6-methyl ester	C_18_H_28_O_8_	^−^H	371.1714	0.8	[58]	other
77	36.43	Naringenin	C_15_H_12_O_5_	^−^H	271.0610	−0.8	[62]	flavones
78	37.25	3, 3′-diisofraxidin	C_22_H_18_O_10_	^+^H	443.0971	−0.5	[27]	coumarin
79	37.37	Arteminorin A	C_22_H_18_O_10_	^−^H	441.0828	0.1	[63]	coumarin
80	37.95	Kaempferol	C_15_H_10_O_6_	^−^H	285.0400	−1.7	RS [49]	flavones
81	38.71	3,7-Dihydroxy-2,4-dimethoxyphenanthrene	C_16_H_14_O_4_	^−^H	269.0827	3.0	[64]	other
82	40.52	Shizukanolide	C_15_H_18_O_3_	^+^H	247.1327	−0.6	[65]	sesquiterpenoids
83	41.85	Pinocembrine	C_15_H_12_O_4_	^−^H	255.0669	2.5	[66]	flavones
84	42.55	Atractylenolide I	C_15_H_18_O_2_	^+^H	231.1378	−0.9	[39]	sesquiterpenoids
85	42.55	Atractylenolide III	C_15_H_20_O_3_	^+^H	249.1484	−0.5	[39]	sesquiterpenoids
86	43.57	Nudol	C_16_H_14_O_4_	^−^H	269.0830	3.9	[67]	phenanthrene
87	43.92	Shizukanolide A	C_15_H_18_O_2_	^+^H	231.1377	−0.9	[68]	sesquiterpenoids
88	44.54	Atractylenolide II	C_15_H_20_O_2_	^+^H	233.1533	−1.3	RS [39]	sesquiterpenoids
89	45.29	Chloranthalactone A	C_15_H_16_O_2_	^+^H	229.1222	−0.7	[69]	sesquiterpenoids
90	46.86	Spathulenol	C_15_H_24_O	^+^H	221.1896	−1.6	[70]	sesquiterpenoids
91	49.02	Linolenic acid	C_18_H_30_O_2_	^−^H	277.2186	4.5	RS [60]	organic acid
92	50.17	Linoleic acid	C_18_H_32_O_2_	^−^H	279.2341	4.0	RS [60]	organic acid

Note: “Basis” indicates compositional identification based on the literature and standards, and “RS” indicates comparison with reference standards.

**Table 2 molecules-30-01825-t002:** Evaluation of similarity in SH herbs’ fingerprints.

Numbers	Similarity	Numbers	Similarity
S1	0.971	S13	0.916
S2	0.922	S14	0.991
S3	0.933	S15	0.882
S4	0.970	S16	0.936
S5	0.947	S17	0.957
S6	0.957	S18	0.945
S7	0.815	S19	0.992
S8	0.960	S20	0.842
S9	0.975	S21	0.903
S10	0.951	S22	0.970
S11	0.863	S23	0.848
S12	0.900	S24	0.881

**Table 3 molecules-30-01825-t003:** Sources and origins of the herbs of SH.

Numbers	Source	Numbers	Source
S1	Yongfu Town, Zhangping City, Longyan City, Fujian Province	S13	Xingan County, Ji’an City, Jiangxi Province
S2	Pucheng County, Nanping City, Fujian Province	S14	Hongya County, Meishan City, Sichuan Province
S3	Tuorong County, Ningde City, Fujian Province	S15	Panzhihua City, Sichuan Province
S4	Yong’an City, Sanming City, Fujian Province	S16	Laochang Township, Ya’an City, Sichuan Province
S5	Nanjing County, Zhangzhou City, Fujian Province	S17	Si Jing Township, Ya’an City, Sichuan Province, China
S6	Liping County, Qiandongnan Prefecture, Guizhou Province	S18	Wenshan Zhuang and Miao Autonomous Prefecture, Yunnan Province
S7	Sandu County, Qiannan Prefecture, Guizhou Province	S19	Tiandeng County, Chongzuo City, Guangxi Zhuang Autonomous Region
S8	Shibing County, Qiandongnan Miao and Dong Autonomous Prefecture, Guizhou Province	S20	Guilin City, Guangxi Zhuang Autonomous Region
S9	Chongyi County, Ganzhou City, Jiangxi Province	S21	Lingchuan County, Guilin City, Guangxi Zhuang Autonomous Region
S10	Dayu County, Ganzhou City, Jiangxi Province	S22	Ziyuan County, Guilin City, Guangxi Zhuang Autonomous Region
S11	Quannan County, Ganzhou City, Jiangxi Province	S23	Babu District, Hezhou City, Guangxi Zhuang Autonomous Region
S12	Anfu County, Ji’an City, Jiangxi Province	S24	Liuzhou City, Guangxi Zhuang Autonomous Region

## Data Availability

Data are reported in the paper.

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
