# Peer review of "Multi-Component Characterization and Quality Evaluation Strategy of Sarcandrae Herba by Combining Dual-Column Tandem HPLC Fingerprint and UPLC-Q-TOF-MS/MS"

_molecules, 2025, doi:10.3390/molecules30081825_

Round 1

Reviewer 1 Report

Comments and Suggestions for Authors

Reviewer’s comments:

At first, the reviewer would like to draw the authors’ attention to the fact that in the text of manuscript there are two Tables with the same № 1 (page 4, line 133, and page 15, line 230). Please renumber the last one to Table № 3.

Of course, Traditional Chinese Medicine (TCM) is full of exotics, and the number of objects to be characterized by contemporary analytical methods is rather large. The herb of Sarcandrae herba is one of them.

To optimize the separation of the constituents of complex mixtures, the “tandem-connected” HPLC-columns was used for their separation. On the one hand, this is an advantage, but on the other hand, it is not. This is because the same combination of columns will need to be used when reproducing the results with application of fingerprinting technique. However, this cannot be considered a fundamental mistake of the authors, because it is the typical fault for fingerprinting techniques based on the absolute chromatographic retention parameters. Data analysis was fulfilled using standard PCA (Principal Component Analysis) algorithms. It was estimated that PCA based on the common components could not realize the distinction of the origin of the herbs, while the integrated analysis based on fingerprint profiles and mass spectrometry could realize the distinction of the origin of the herbs (principal authors’ conclusion).

The most important part of the manuscript is Table 1 with the results of the identification of 93 constituents of Sarcandrae herba extracts. The key question concerning this Table is the identification of compounds. As we can conclude, it is based on the previous mentions of all compound in the literature (references are provided) in combination with high resolution mass spectrometric information for ions [M – H]-. Obviously, this fact (backgrounds of identification) deserves more detailed authors’ comments.

Additionally, regarding Table 1, it is necessary to explain, why most of m/z values are indicated with four decimal digits, but some of them (e.g., 6, 8, 21, 23, 25, etc.) with three decimal digits only. The reference for previous determination is absent for compound № 83, namely 1,2-bis(4-methylphenyl)-1,2-ethanedione. So far as this structure is rather surprising for natural compounds, the reasons for that should be explained in details, as well.

All mentioned above means the necessity of any (small or moderate) revision of the manuscript.

Reviewer 2 Report

Comments and Suggestions for Authors

Sarcandrae herba is a versatile herb with a wide range of applications in traditional medicine. Its anti-inflammatory, immune-boosting, and antioxidant properties make it a valuable natural remedy for various health conditions. As research into its bioactive compounds continues to grow, Sarcandrae herba holds promise for broader applications in modern medicine while maintaining its roots in traditional healing practices. The authors have successfully demonstrated that dual-column tandem HPLC enables a more comprehensive analysis of *Sarcandrae herba* extracts, leading to the identification of a significant number of analytes. Based on these findings, I find the manuscript to be worthy of publication. However, there are a few areas that require additional information and further discussion:

  1. Please provide more details regarding the IDA mode, including the accumulation times, the number of selected peaks, and the CE values.
  2. Include a clear description of the procedures used for compound identification to enhance the reproducibility and clarity of the study.
  3. The term "ion flow" should be replaced with "ion current" for accuracy and consistency in terminology.

Reviewer 3 Report

Comments and Suggestions for Authors

The authors present a manuscript that deals with the analysis of Sarcandrae herba by combining dual column tandem HPLC fingerprint and UPLC-Q-TOF-MS.

I think this is an interesting paper and the authors identified some unique compounds. But I have a few comments

Abstract - The last sentence seems contradictory? Maybe clarify whether you were able to differentiate the origins?  "The results showed that the principal component analysis based on common components could not realize the discrimination between the origins, and the overall analysis based on fingerprinting and mass spectrometry could differentiate the origins of the herbs."

Introduction - generally well written an d referenced. I found the justification for the study slightly too long in terms of the current processes for quality assessment etc. 

I think the aims of objectives for the study in the last paragraph could be more specific about the origins and also techniques etc. The end sentence seems to give a conclusion to the study which I'm not sure should be in the Introduction?

"and had far reaching significance in ensuring the stability of the therapeutic efficacy of the SH related products. "

Results and discussion 

This sentence could have a reference? "The traditional liquid chromatographic separation process usually selects a single
column for HPLC separation, but the separation of components is not effective."

Could be written more scientifically "Figure 1, and it can be seen that compared with the single-column mode, the fingerprints established by the dual-column tandem mode had more signal peaks, and the separation degree also showed better performance"

Figure 1 needs a better figure legend - it does not give any useful scientific information - columns/conditions etc.

You use the terms "different origins", "six provinces" etc. but can you just state the origins of the herbs and refer to Table1?

It is not easy to interpret Figure 2 and it needs a better figure legend i.e. sample information etc.

Figure 3. also could have a more specific figure legend - they are not diagrams for example?

Table 1 is quite large currently - what is the column labelled "basis"? that contains a reference. Maybe needs further explanation.

This section seems like methodology?

"The test results of the samples of each origin were imported into the similarity evaluation system software “Chinese Medicine Chromatographic Fingerprint Similarity Evaluation System (Version 2012.130723)”, and the sample S1 was selected as the reference fingerprint, and the median method was used as the method of generating the control fingerprint, and the multi-point correction method was combined to match the liquid phase profiles of each origin."

Figure 4 says "Reference fingerprint (characterization) of SH herbs" - how is this established?

Figure 5 doesn't work that well as it is not easy to see the colour differences in the circles - maybe use different shaped markers for different provinces?

Some information repeated - ", most of which have antibacterial, anti-inflammatory, anti-platelet aggregation, anti-disease antibacterial, anti-inflammatory, anti-platelet aggregation, anti-toxicity and other biological activitie"

Isn't clearly written " It demonstrated that the HPLC fingerprints can components reflect the overall characteristics and differences of the herbs, which is beneficial to the identification of the herbs' superiority and origin"

This also seems like methodology 

"As a further means of comparing the differences between the components in SH samples from different origins, the sample information is presented in Figure 7 as a fingerprint, which normalizes the signal peaks for each substance from the different origins, transforming them into the range from 0 to 1. The fingerprint contains all the signals that can be detected by the instrument, where each row represents a sample and each column represents a substance. In Fig. 7, the substances in regions I, II, III, IV and V represent organic acids, sesquiterpenoids, flavones, coumarin and other classes of characteristic components, respectively, and the differences in compositional composition of SH herbs from different origins can be observed distinctly. "

Methodology - this needs more specific information "The extract was kept cool and 241
weighed again, then supplemented with 60% methanol, mixed well and filtered." Temperatures? volumes etc. check throughout method that all parameters are outlined.

Needs more information "Appropriate amount of each reference substance was weighed precisely, placed in a 50 mL brown volumetric flask, added with methanol and dissolved by ultrasonication and volume, and then passed through 0.22μm microporous filter membrane to obtain the standard solutions." Also why is it filtered?

Conclusion - I think the conclusion should sum up the future directions i.e. now that a method has been established that indicates differences in the herb products what is the use of this - I think the authors mentioned a medicinal study based on the origins of the different herbs i.e. establishing effectiveness in treatment etc.?

Round 2

Reviewer 2 Report

Comments and Suggestions for Authors

The authors have addressed all issues and the manuscript can be published in my opinion.

Reviewer 3 Report

Comments and Suggestions for Authors

The authors did a good job of addressing my comments and I think the current manuscript is imroved and could be published in molecules